# Transferring to Real-World Layouts:
# A Depth-aware Framework for Scene Adaptation

Submission Id: 269

## ABSTRACT

Scene segmentation via unsupervised domain adaptation (UDA) enables the transfer of knowledge acquired from source synthetic data to real-world target data, which largely reduces the need for manual pixel-level annotations in the target domain. To facilitate domain-invariant feature learning, existing methods typically mix data from both the source domain and target domain by simply copying and pasting the pixels. Such vanilla methods are usually sub-optimal since they do not take into account how well the mixed layouts correspond to real-world scenarios. Real-world scenarios are with an inherent layout. We observe that semantic categories, such as sidewalks, buildings, and sky, display relatively consistent depth distributions, and could be clearly distinguished in a depth map. Based on such observation, we propose a depth-aware framework to explicitly leverage depth estimation to mix the categories and facilitate the two complementary tasks, *i.e.*, segmentation and depth learning in an end-to-end manner. In particular, the framework contains a Depth-guided Contextual Filter (DCF) for data augmentation and a cross-task encoder for contextual learning. DCF simulates the real-world layouts, while the cross-task encoder further adaptively fuses the complementing features between two tasks. Besides, it is worth noting that several public datasets do not provide depth annotation. Therefore, we leverage the off-the-shelf depth estimation network to generate the pseudo depth. Extensive experiments show that our proposed methods, even with pseudo depth, achieve competitive performance on two widely-used benchmarks, *i.e.*, 77.7 mIoU on GTA→Cityscapes and 69.3 mIoU on Synthia→Cityscapes.

## CCS CONCEPTS

• **Computing methodologies → Scene understanding**; **Transfer Learning**.

## KEYWORDS

Unsupervised Scene Adaptation, Depth-aware Fusion, Transfer Learning, Self-supervised Learning

## 1 INTRODUCTION

Semantic segmentation refers to the task of assigning pixel-level category labels in an image, which has achieved significant progress in the last few years [2, 6, 35, 63]. It is worth noting that prevailing

*ACM Multimedia '24, 28 October 2024 - 1 November 2024, Melbourne, Australia*
© 2024 Association for Computing Machinery.
ACM ISBN 978-x-xxxx-xxxx-x/YY/MM. . . $15.00
https://doi.org/10.1145/nnnnnnn.nnnnnnn

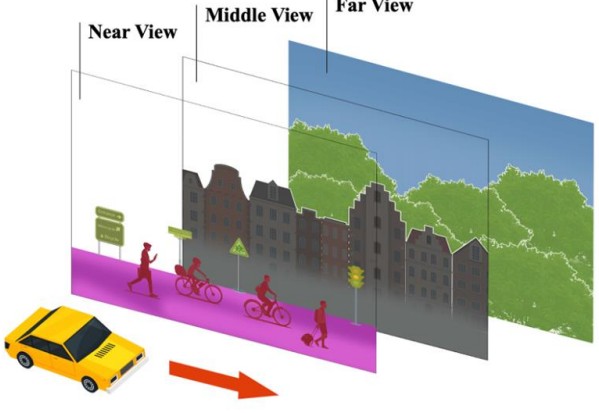

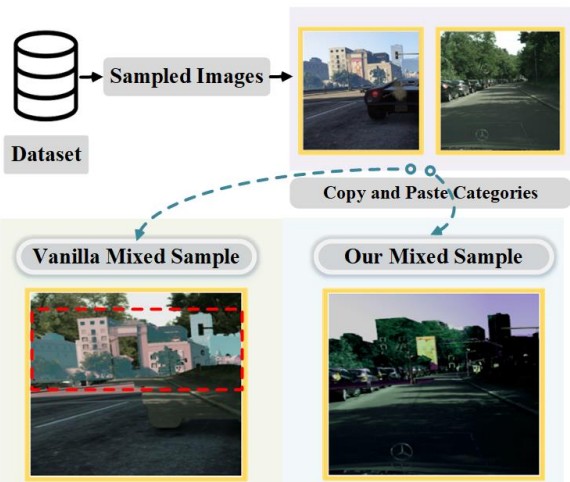

**Figure 1: (a) Considering the driving scenario, we observe that the object location is relatively stable according to the distance from the camera. Therefore, we propose a Depth-guided Contextual Filter (DCF) which is aware of the semantic categories distribution in terms of Near, Middle, and Far view to facilitate cross-domain mixing. (b) Since we explicitly take the semantic layout into consideration, our method achieves more realistic mixed samples compared to the competitive MIC (Vanilla Mixed Sample) [20]. As shown in the red dotted box, "new" buildings are pasted before the parked cars.**

models usually require large-scale training datasets with high-quality annotations, such as ADE20K [78], to achieve good performance and but such pixel-level annotations in real-world are usually unaffordable and time-consuming [11]. One straightforward idea is to train networks with synthetic data so that the pixel-level annotations are easier to obtain [43, 44]. However, the network trained

with synthetic data usually results in poor scalability when being deployed to a real-world environment due to multiple factors, such as weather, illumination, and road design. Therefore, researchers resort to unsupervised domain adaptation (UDA) to further tackle the variance between domains. One branch of UDA methods attempts to mitigate the domain shift by aligning the domain distributions [17, 37, 42, 51, 61]. Another potential paradigm to heal the domain shift is self-training [30, 67, 76, 81, 82], which recursively refine the target pseudo-labels. Taking one step further, recent DACS [50] and follow-up works [8, 18–20, 23, 57, 62] combine self-training and ClassMix [40] to mix images from both source and target domain. In this way, these works could craft highly perturbed samples to assist training by facilitating learning shared knowledge between two domains. Specifically, cross-domain mixing aims to copy the corresponding regions of certain categories from a source domain image and paste them onto an unlabelled target domain image. We note that such a vanilla strategy leads to pasting a large amount of objects to the unrealistic depth position. It is because that every category has its own position distribution. For instance, the background classes such as "sky" and "vegetation" usually appear farther away, while the classes that occupy a small number of pixels such as "traffic signs" and "pole", usually appear closer as shown in Figure 1 (a). Such crafted training data compromise contextual learning, leading to sub-optimal location prediction performance, especially for small objects.

To address these limitations, we observe the real-world depth distribution and find that semantic categories are easily separated (disentangled) in the depth map since they follow a similar distribution under certain scenarios, *e.g.*, urban. Therefore, we propose a new depth-aware framework, which contains Depth Contextual Filter (DCF) and a cross-task encoder. In particular, DCF removes unrealistic classes mixed with the real-world target training samples based on the depth information. On the other hand, multi-modal data could improve the performance of deep representations and the effective use of the deep multi-task features to facilitate the final predictions is crucial. The proposed cross-task encoder contains two specific heads to generate intermediate features for each task and an Adaptive Feature Optimization module (AFO). AFO encourages the network to optimize the fused multi-task features in an end-to-end manner. Specifically, the proposed AFO adopts a series of transformer blocks to capture the information that is crucial to distinguish different categories and assigns high weights to discriminative features and vice versa.

The main contributions are as follows: (**1**) We propose a simple Depth-Guided Contextual Filter (DCF) to explicitly leverage the key semantic categories distribution hidden in the depth map, enhancing the realism of cross-domain information mixing and refining the cross-domain layout mixing. (**2**) We propose an Adaptive Feature Optimization module (AFO) that enables the cross-task encoder to exploit the discriminative depth information and embed it with the visual feature which jointly facilitates semantic segmentation and pseudo depth estimation. (**3**) Albeit simple, the effectiveness of our proposed methods has been verified by extensive ablation studies. Despite the pseudo depth, our method still achieves competitive accuracy on two commonly used scene adaptation benchmarks, namely 77.7 mIoU on GTA→Cityscapes and 69.3 mIoU on Synthia→Cityscapes.

## 2 RELATED WORK

### 2.1 Unsupervised Domain Adaptation

Unsupervised domain adaptation (UDA) aims to train a model on a label-rich source domain and adapt the model to a label-scarce target domain. Some methods propose learning the domain-invariant knowledge by aligning the source and target distribution at different levels. For instance, AdaptSegNet [51], ADVENT [54], and CLAN [37] adversarially align the distributions in the feature space. CyCADA [17] diminishes the domain shift at both pixel-level and feature-level representation. DALN [4] proposes a discriminator-free adversarial learning network and leverages the predicted discriminative information for feature alignment. Both Wu *et al.*[61] and Yue *et al.* [68] learn domain-invariant features by transferring the input images into different styles, such as rainy and foggy, while Zhao *et al.* [74] and Zhang *et al.* [71] diversify the feature distribution via normalization and adding noise respectively. Another line of work refines pseudo-labels gradually under the iterative self-training framework, yielding competitive results. Following the motivation of generating highly reliable pseudo labels for further model optimization, CBST [81] adopts class-specific thresholds on top of self-training to improve the generated labels. Feng *et al.*[12] acquire pseudo labels with high precision by leveraging the group information. PyCDA [32] constructs pseudo-labels in various scales to further improve the training. Zheng *et al.*[75] introduce memory regularization to generate consistent pseudo labels. Other works propose either confidence regularization [76, 82] or category-aware rectification [69, 70] to improve the quality of pseudo labels. DACS [50] proposes a domain-mixed self-training pipeline to mix cross-domain images during training, avoiding training instabilities. Kim *et al.*[25], Li *et al.*[31] and Wang *et al.*[56] combine adversarial and self-training for further improvement. Chen *et al.*[5] establish a deliberated domain bridging (DDB) that aligns and interacts with the source and target domain in the intermediate space. SePiCo [62] and PiPa [8] adopt contrastive learning to align the domains. Liu *et al.*[34] addresses the label shift problem by adopting class-level feature alignment for conditional distribution alignment. Researchers also attempted to perform entropy minimization [7, 54], and image translation [15, 65]. consistency regularization[1, 10, 39, 79]. Recent multi-target domain adaptation (MTDA) methods enable a single model to adapt a labeled source domain to multiple unlabeled target domains [13, 28, 47]. However, the above methods usually ignore the rich multi-modality information, which can be easily obtained from the depth sensor and other sensors.

### 2.2 Depth Estimation and Multi-task Learning in Semantic Segmentation

Semantic segmentation and geometric information are shown to be highly correlated [24, 49, 53, 58, 64, 72, 73]. Recently depth estimation has been increasingly used to improve the learning of semantics within the context of multi-task learning, but the depth information should be exploited more precisely to help the domain adaptation. SPIGAN [27] pioneered the use of geometric information as an additional supervision by regularizing the generator with an auxiliary depth regression task. DADA [55] introduces an adversarial training framework based on the fusion of semantic and depth predictions to facilitate the adaptation. GIO-Ada [9] leverages

the geometric information on both the input level and output level to reduce domain shift. CTRL [45] encodes task dependencies between the semantic and depth predictions to capture the cross-task relationships. CorDA [57] bridges the domain gap by utilizing self-supervised depth estimation on both domains. Wu *et al.* [60] propose to further support semantic segmentation by depth distribution density. Our work follows a similar spirit to leverage depth knowledge as auxiliary supervision. It is worth noting that our work is primarily different from existing works in the following two aspects: (1) from the data perspective, we explicitly delineate the depth distribution to refine data augmentation and construct realistic training samples to enhance contextual learning. (2) from the network perspective, our proposed multi-task learning network not only adopts auxiliary supervision for learning more robust deep representations but also facilitates the multi-task feature fusion by iterative deploying of transformer blocks to jointly learn the rich multi-task information for improving the final predictions.

## 3 METHOD

### 3.1 Problem Formulation

In a typical Unsupervised Domain Adaptation (UDA) scenario, we have a source domain, denoted $S$, which consists of abundant labeled synthetic data. On the other hand, the target domain, represented by $T$, contains unlabeled real-world data. For example, we have labeled training samples $\left(\mathbf{x}^S, \mathbf{y}^S, \mathbf{z}^S \sim \mathbf{X}^S, \mathbf{Y}^S, \mathbf{Z}^S\right)$ in the source domain, where $\mathbf{x}^S, \mathbf{y}^S$ are the training image and the corresponding ground truth for semantic segmentation. $\mathbf{z}^S$ is the label for the depth estimation task. Similarly, we have unlabeled target images sampled from target domain data $\left(\mathbf{x}^T, \mathbf{z}^T \sim \mathbf{X}^T, \mathbf{Z}^T\right)$, where $\mathbf{x}^T$ is the unlabeled sample in the target domain and $\mathbf{z}^T$ is the label for the depth estimation task. Since depth annotation is not supported by common public datasets, we adopt pseudo depth that can be easily generated by the off-the-shelf model [14].

### 3.2 Depth-guided Contextual Filter

In UDA, recent works Recent UDA works [8, 18–20, 40, 57] often employ pixel mixing to create cross-domain augmented samples. The basic idea is straightforward: take a portion of pixels from a source domain image and transplant them onto an equivalent area in a target domain image. However, this simple approach faces challenges due to the inherent differences in structure and layout between source and target domain data. To decrease noisy signals and simulate augmented training samples with real-world layouts, we propose Depth-guided Contextual Filter (DCF) to reduce the noisy pixels that are naively mixed across domains. The implementation of DCF is represented as pseudo-code in Algorithm 1, where the image $\mathbf{x}^S$ and the corresponding semantic labels $\mathbf{y}^S$ are sampled from source domain data. The image $\mathbf{x}^T$ and the depth label $\mathbf{z}^T$ are from target domain data. Pseudo label $\hat{\mathbf{y}}^T$ is then generated as $\hat{\mathbf{y}}^T = \mathcal{F}_\theta\left(\mathbf{x}^T\right)$, where $\mathcal{F}_\theta$ is a pre-trained semantic network. In practice, $\mathcal{F}_\theta$ usually has been trained on the source domain dataset via supervised learning. Based on the hypothesis that most semantic categories usually fall under a finite depth range, we introduce DCF, which divides the target depth map $\mathbf{z}^T$ into a few discrete depth

---

**Algorithm 1** Depth-guided Contextual Filter Algorithm with Cross-Image Mixing and Self Training

---

**Input:** Source domain: $(\mathbf{x}^S, \mathbf{y}^S, \mathbf{z}^S \sim \mathbf{X}^S, \mathbf{Y}^S, \mathbf{Z}^S)$, Target domain: $(\mathbf{x}^T, \mathbf{z}^T \sim \mathbf{X}^T, \mathbf{Z}^T)$. Semantic network $\mathcal{F}_\theta$.

1: Initialize network parameters $\theta$ randomly.
2: **for** iteration = 1 to $n$ **do**
3:   $\hat{\mathbf{y}}^T \leftarrow \mathcal{F}_\theta\left(\mathbf{x}^T\right)$, Generate pseudo label
4:   Pre-calculate the density value $\mathbf{p}$ for each class $i$ at each depth interval from the target depth map $\mathbf{z}^T$,
5:   $\hat{\mathbf{y}}^M \leftarrow \mathcal{M} \odot \mathbf{y}^S + (1 - \mathcal{M}) \odot \hat{\mathbf{y}}^T$, Randomly select 50% categories and copy the category ground truth label from the source image to target pseudo label
    $\mathbf{x}^M \leftarrow \mathcal{M} \odot \mathbf{x}^S + (1 - \mathcal{M}) \odot \mathbf{x}^T$, Copy the corresponding category region from the source image to the target image
6:   Re-calculate the density value $\hat{\mathbf{p}}$ after the mixing,
7:   Calculate the depth density distribution difference before and after mixing,
8:   Filter the category once the difference exceeds the threshold,
9:   Re-generate the depth-aware binary mask $\mathcal{M}^{DCF}$,
10:  $\hat{\mathbf{y}}^F \leftarrow \mathcal{M}^{DCF} \odot \mathbf{y}^S + \left(1 - \mathcal{M}^{DCF}\right) \odot \hat{\mathbf{y}}^T$, Generate the filtered training samples with new DCF mask
    $\mathbf{x}^F \leftarrow \mathcal{M}^{DCF} \odot \mathbf{x}^S + \left(1 - \mathcal{M}^{DCF}\right) \odot \mathbf{x}^T$,
11:  Compute predictions
    $\bar{\mathbf{y}}^S \leftarrow argmax\left(\mathcal{F}_\theta\left(\mathbf{x}^S\right)\right)$,
    $\bar{\mathbf{y}}^F \leftarrow argmax\left(\mathcal{F}_\theta\left(\mathbf{x}^F\right)\right)$,
12:  Compute loss for the batch:
    $\ell \leftarrow \mathcal{L}\left(\bar{\mathbf{y}}^S, \mathbf{y}^S, \bar{\mathbf{y}}^F, \hat{\mathbf{y}}^F\right)$.
13:  Compute $\nabla_\theta \ell$ by backpropagation.
14:  Perform stochastic gradient descent.
15: **end for**
16: **return** $\mathcal{F}_\theta$

---

intervals $(\Delta z_1, ..., \Delta z_n)$. For a given real-world target input image $\mathbf{x}^T$ combined with the pseudo label $\hat{\mathbf{y}}^T$ and target depth map $\mathbf{z}^T$, the density value at each depth interval $(\Delta z_1, ..., \Delta z_n)$ for each class $i \in (1, \ldots, C)$ can be counted and normalized as a probability. We denote the density value for class $i$ at the depth interval $\Delta z_1$ as $p_i(\Delta z_1)$. All the density values make up the depth distribution in the target domain image. Then we randomly select half of the categories on the source images to paste on the target domain image. In practice, we apply a binary mask $\mathcal{M}$ to denote the corresponding pixels. Then naive cross-domain mixed image $\mathbf{x}^{Mix}$ and the mixed label $\hat{\mathbf{y}}^{Mix}$ can be formulated as:

$$\mathbf{x}^{Mix} = \mathcal{M} \odot \mathbf{x}^S + (1 - \mathcal{M}) \odot \mathbf{x}^T, \tag{1}$$

$$\hat{\mathbf{y}}^{Mix} = \mathcal{M} \odot \mathbf{y}^S + (1 - \mathcal{M}) \odot \hat{\mathbf{y}}^T, \tag{2}$$

where $\odot$ denotes the element-wise multiplication of between the mask and the image. The naively mixed images are visualized in Figure 2. It could be observed that due to the depth distribution difference between two domains, pixels of "Building" category are mixed from the source domain to the target domain, creating unrealistic images. Training with such training samples will compromise contextual learning. Therefore, we propose to filter the pixels that

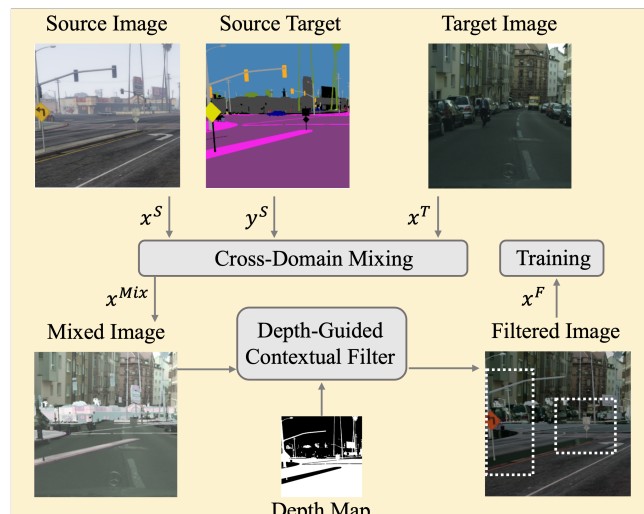

**Figure 2: Source domain images $x^S$ and $x^T$ are mixed together, using the ground truth label $y^S$. The mixed images are de-noised by our proposed Depth-guided Contextual Filter (DCF) and then trained by the network. We illustrate DCF with a set of practical sample. As illustrated, the unrealistic "Building" pixels from the source image are mixed pasted to the target image, leading to a noisy mixed sample. The proposed DCF removes these pixels and maintain mixed pixels of "Traffic Sign" and "Pole" shown in the white dotted boxes, enhancing the realism of cross-domain mixing. (Best viewed when zooming in.)**

do not match the depth density distribution in the mixed image. After the naive mixing, we re-calculate the density value for each class at each depth interval. For example, the new density value for class $i$ at the depth interval $\Delta z_1$ is denoted as $\hat{p}_i(\Delta z_1)$. Then we calculate the depth density distribution difference for each pasted category and denote the difference for class $i$ at the depth interval $\Delta z_1$ as $\Delta p_i(\Delta z_1) = |p_i(\Delta z_1) - \hat{p}_i(\Delta z_1)|$. Once $\Delta p_i(\Delta z_1)$ exceeds the threshold of that category $i$, these pasted pixels are removed. After performing DCF, we confirm the final realistic pixels to be mixed and construct a depth-aware binary mask $\mathcal{M}^{\mathcal{DCF}}$, which is changed dynamically based on the depth layout of the current target image.

The filtered mixing samples are then generated. In practice, we directly apply the updated depth-aware mask to replace the original mask. Therefore, the new mixed sample and the label are as follows:

$$\mathbf{x}^F = \mathcal{M}^{\mathcal{DCF}} \odot \mathbf{x}^S + \left(1 - \mathcal{M}^{\mathcal{DCF}}\right) \odot \mathbf{x}^T, \qquad (3)$$

$$\hat{\mathbf{y}}^F = \mathcal{M}^{\mathcal{DCF}} \odot \mathbf{y}^S + \left(1 - \mathcal{M}^{\mathcal{DCF}}\right) \odot \hat{\mathbf{y}}^T. \qquad (4)$$

Because large objects such as "sky" and "terrain" usually aggregate and occupy a large amount of pixels and small objects only occupy a small amount of pixels in a certain depth range, we set different filtering thresholds for each category. DCF uses pseudo semantic labels for the target domain as there is no ground truth available. Since the label prediction is not stable in the early stage, we apply a warmup strategy to perform DCF after 10,000 iterations. Examples of the input images, naively mixed samples and filtered samples

are presented in Figure 2. The sample after the process of the DCF module has the pixels from the source domain that match the depth distribution of the target domain, helping the network to better deal with the domain gap.

### 3.3 Multi-task Scene Adaptation Framework

In order to exploit the relation between segmentation and depth learning, we introduce a multi-task scene adaptation framework including a high resolution semantic encoder, and a cross-task shared encoder with a feature optimization module, which is depicted in Figure 3. The proposed framework incorporates and optimizes the fusion of depth information for improving the final semantic predictions.

*High Resolution Semantic Prediction.* Most supervised methods use high resolution images for training, but common scene adaptation methods usually use random crops of the image that is half of the full resolution. To reduce the domain gap between scene adaptation and supervised learning while maintaining the GPU memory consumption, we adopt a high-resolution encoder to encode HR image crops into deep HR features. Then a semantic decoder is used to generate the HR semantic predictions $\bar{\mathbf{y}}_{hr}$. We adopt the cross entropy loss for semantic segmentation:

$$\mathcal{L}_{hr}^S\left(\mathbf{x}^S, \mathbf{y}^S\right) = \mathbb{E}\left[-\mathbf{y}^S \log \bar{\mathbf{y}}_{hr}^S\right], \qquad (5)$$

$$\mathcal{L}_{hr}^F\left(\mathbf{x}^F, \mathbf{y}^F\right) = \mathbb{E}\left[-\hat{\mathbf{y}}^F \log \bar{\mathbf{y}}_{hr}^F\right], \qquad (6)$$

where $\bar{\mathbf{y}}_{hr}^S$ and $\bar{\mathbf{y}}_{hr}^T$ are high resolution semantic predictions. $\mathbf{y}^S$ is the one-hot semantic label for the source domain and $\hat{\mathbf{y}}^F$ is the one-hot pseudo label for the depth-aware fused domain.

*Adaptive Feature Optimization.* In addition to the high resolution encoder, We use another cross-task encoder to encode input images which are shared for both tasks. Depth maps are rich in spatial depth information, but a naive concatenation of depth information directly to visual information causes some interference, e.g. categories at similar depth positions are already well distinguished by visual information, and attention mechanisms can help the network to select the crucial part of the multitask information. In the proposed multi-task learning framework, the visual semantic feature and depth feature is generated by a visual head and a depth head, respectively. As shown in Figure 3, after applying batch normalization, an Adaptive Feature Optimization module then concatenates the normalized input visual feature and the input depth feature to create a fused multi-task feature by concatenation as $f_{fuse}^{in} = \text{CONCAT}\left(f_{vis}^{in}, f_{depth}^{in}\right)$. The fused feature is then fed into a series of transformer blocks to capture the key information between the two tasks. The attention mechanism adaptively adjusts the extent to which depth features are embedded in visual features:

$$f_{fuse}^{out} = \mathcal{W}_{Trans}\left(f_{fuse}^{in}\right), \qquad (7)$$

where $\mathcal{W}_{Trans}$ is the transformer parameter. The learned output of the transformer blocks is a weight map $\gamma$ which is multiplied back to the input visual feature and depth feature resulting in an optimized feature as:

$$\gamma = \sigma\left(\mathcal{W}_{Conv} \otimes f_{fuse}^{out}\right), \qquad (8)$$

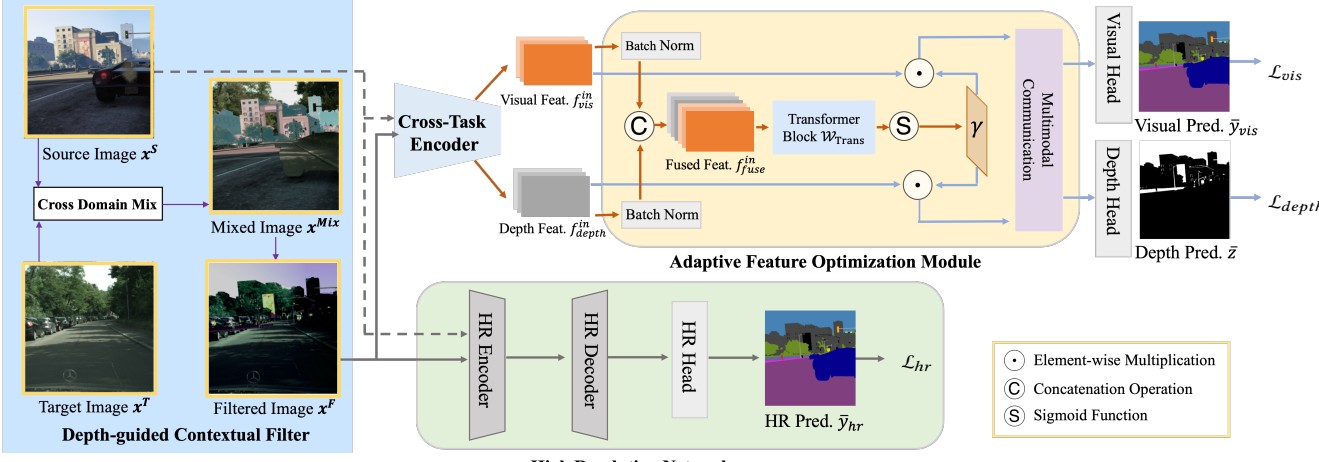

**Figure 3: The proposed multi-task learning framework. The input images $x^F$ are mixed from the source image $x^S$ and target domain $x^T$ according to the depth (Please refer to Figure 2). Then we are fed $x^S$ and $x^F$ into the high resolution encoder to generate high resolution predictions. To enhance multi-modal learning, the visual and depth feature created by the cross-task encoder are fused and fed into the proposed Adaptive Feature Optimization module (AFO) for multimodal communication. Finally, the multimodal communication via several transformer blocks incorporates and optimizes the fusion of depth information, improving the final visual predictions.**

where $\mathcal{W}_{Conv}$ denotes the convolution parameter, $\otimes$ denotes the convolution operation and $\sigma$ represents the sigmoid function. The weight matrix $\gamma$ performs adaptive optimization of the muti-task features. Then, the fused feature $f_{fuse}^{out}$ is fed into different decoders for predicting different final tasks, *i.e.*, the visual and the depth task. The output features are essentially multimodal features containing crucial depth information:

$$f_{vis}^{out} = f_{vis}^{in} \odot \gamma, \quad f_{depth}^{out} = f_{depth}^{in} \odot \gamma, \tag{9}$$

where $\odot$ represents element-wise multiplication. The optimized visual and depth feature is then fed into the multimodal communication module for further processing. The multimodal communication module refines the learning of key information between two tasks by iterative use of transformer blocks. the inference is merely based on the visual input when the feature optimization is fished. The final semantic prediction $\bar{y}_{vis}^S$ and depth prediction $\bar{z}^S$ can be generated from the final visual feature $f_{vis}^{final}$ and depth feature $f_{depth}^{final}$ by the visual head and depth head . Similar to the high resolution predictions, we use the cross entropy loss for the semantic loss calculation:

$$\mathcal{L}_{vis}^S \left( \mathbf{x}^S, \mathbf{y}^S \right) = \mathbb{E} \left[ -\mathbf{y}^S \log \bar{\mathbf{y}}_{vis}^S \right], \tag{10}$$

$$\mathcal{L}_{vis}^F \left( \mathbf{x}^F, \mathbf{y}^F \right) = \mathbb{E} \left[ -\hat{\mathbf{y}}^F \log \bar{\mathbf{y}}_{vis}^F \right]. \tag{11}$$

We also employ the berHu loss for depth regression at source domain:

$$\mathcal{L}_{depth}^S \left( \mathbf{z}^S \right) = \mathbb{E} \left[ \text{berHu} \left( \bar{\mathbf{z}}^S - \mathbf{z}^S \right) \right], \tag{12}$$

where $\bar{z}$ and $z$ are predicted and ground truth semantic maps. Following [45, 55], we deploy the reversed Huber criterion [26], which

is defined as :

$$\text{ber}\,Hu \left( e_z \right) = \begin{cases} |e_z|, & |e_z| \le H \\ \frac{(e_z)^2 + H^2}{2H}, & |e_z| > H \end{cases} \tag{13}$$
$$H = 0.2 \max \left( |e_z| \right),$$

where $H$ is a positive threshold and we set it to 0.2 of the maximum depth residual. Finally, the overall loss function is:

$$\mathcal{L} = \mathcal{L}_{hr}^S + \mathcal{L}_{vis}^S + \lambda_{depth} \mathcal{L}_{depth}^S + \mathcal{L}_{hr}^F + \mathcal{L}_{vis}^F, \tag{14}$$

where hyperparameter $\lambda_{depth}$ is the loss weight. Considering that our main task is semantic segmentation and the depth estimation is the auxiliary task, we empirically $\lambda_{depth} = 0.1 \times 10^{-2}$. We also designed the ablation studies to change the weight of depth task $\lambda_{depth}$ to the level of $10^{-1}$ or $10^{-3}$.

## 4 EXPERIMENT

### 4.1 Implementation Details

**Datasets.** We evaluate the proposed framework on two scene adaptation settings, *i.e.*, GTA $\rightarrow$ Cityscapes and SYNTHIA $\rightarrow$ Cityscapes, following common protocols [1, 18–20, 50, 57]. Particularly, the GTA5 dataset [43] is the synthetic dataset collected from a video game, which contains 24,966 images annotated by 19 classes. Following [57], we adopt depth information generated by Monodepth2 [14] model which is trained merely on GTA image sequences. SYN-THIA [44] is a synthetic urban scene dataset with 9,400 training images and 16 classes. Simulated depth information provided by SYNTHIA is adopted. GTA and SYNTHIA serve as source domain datasets. The target domain dataset is Cityscapes, which is collected from real-world street-view images. Cityscapes contains 2,975 unlabeled training images and 500 validation images. The resolution of Cityscapes is $2048 \times 1024$ and the common protocol downscales the size to $1024 \times 512$ to save memory. Following [57], the stereo depth

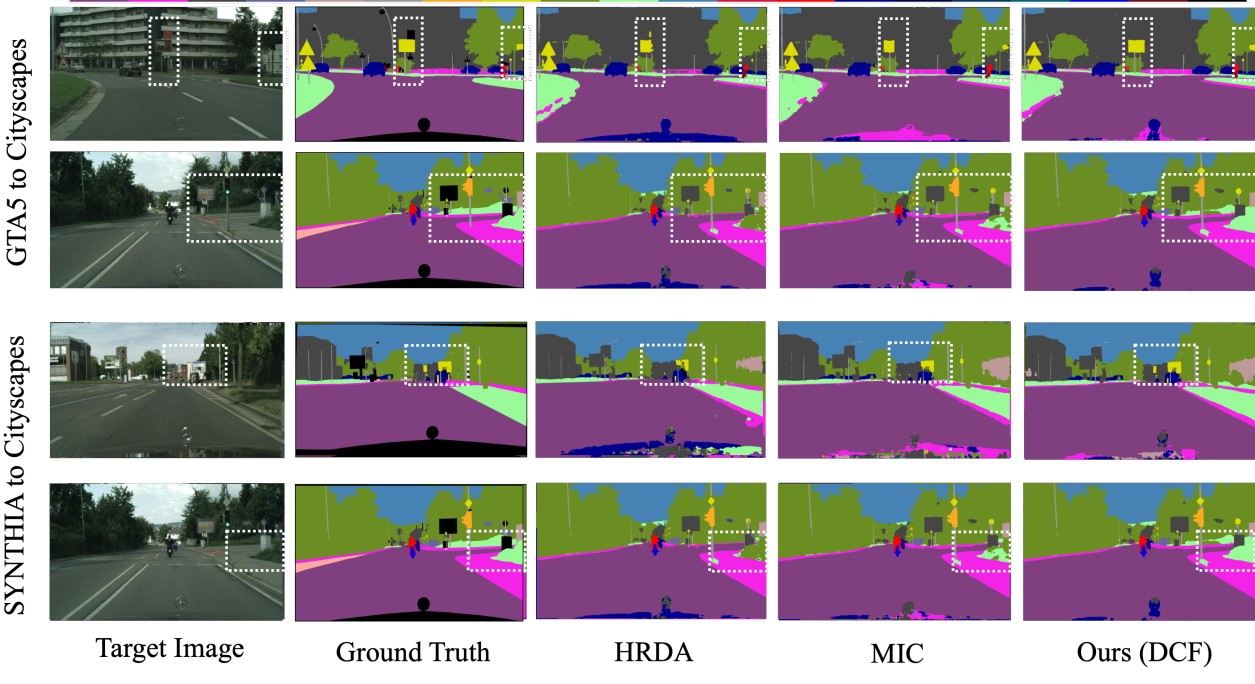

**Figure 4: Qualitative results on GTA → Cityscapes. From left to right: Target Image, Ground Truth, the visual results predicted by HRDA, MIC and Ours. We highlight prediction differences in white dash boxes. The proposed method could predict clear edges.**

estimation from [46] is used. We leverage the Intersection Over Union (IoU) for per-class performance and the mean Intersection over Union (mIoU) over all classes to report the result. The code is based on Pytorch [41]. **We will make our code open-source for reproducing all results.**

**Experimental Setup.** We adopt DAFormer [18] network with MiT-B5 backbone [63] for the high resolution encoder and DeepLabV2 network with ResNet-101 backbone for the cross-task encoder to reduce the memory consumption. All backbones are initialized with ImageNet pretraining. Our training procedure is based on self-training methods with cross-domain mixing [18–20, 50] and enhanced by our proposed Depth-guided Contextual Filter. Following [19, 50], the input image resolution is half of the full resolution for the cross-task encoder and full resolution for high resolution encoder. We utilize the same data augmentation, *e.g.*, color jitter and Gaussian blur and empirically set pseudo labels threshold 0.968 following [50]. We train the network with batch size 2 for 40k iterations on a Tesla V100 GPU.

**Data Resolution.** Our proposed depth-aware multi-task framework contains a high resolution encoder and a cross-task encoder with an adaptive feature optimization module (AFO). Previous works [31, 50, 52] downsample Cityscapes to $1024 \times$ and GTA to $1280 \times 720$. Following [19], for the high resolution encoder, we resize GTA to $2560 \times 1440$ and SYNTHIA to 2560 x 1520. Then the crop size is $1024 \times 1024$. In addition, SegFormer [63] MLP decoder with an embedding dimension of 256 is used for the high resolution branch.

For the cross-task encoder branch, we follow common UDA methods [18, 50] to adopt $1024 \times 512$ pixels (half of the full resolution) for Cityscapes, $1280 \times 760$ for SYNTHIA and $1280 \times 720$ for GTA. In addition, a $512 \times 512$ random crop is extracted.

## 4.2 Comparison with SOTA

**Results on GTA→Cityscapes.** We show our results on GTA → Cityscapes in Table 1 and highlight the best results in bold. It could be observed that our method yields significant performance improvement over the state-of-the-art method MIC [20] from 75.9 mIoU to 77.7 mIoU. Usually, classes that occupy a small number of pixels are difficult to adapt and have a comparably low IoU performance. However, our method demonstrates competitive IoU improvement on most categories especially on small objects such as +5.7 on "Rider", +5.4 on "Fence", +5.2 on "Wall", +4.4 on "Traffic Sign" and +3.4 on "Pole". The result shows the effectiveness of the proposed contextual filter and cross-task learning framework in the contextual learning. Our method also increases the mIoU performance of classes that aggregate and occupy a large amount of pixels in an image by a smaller margin such as +1.8 on "Pedestrain" and +1.1 on "Bike", probably because the rich texture and color information contained in the visual feature already has the ability to recognize these relatively easier classes. The above observations are also qualitatively reflected in Figure 4, where we visualize the segmentation results of the proposed method and the comparison with previous strong transformer-based methods HRDA [19], and MIC [20]. The qualitative results highlighted by white dash boxes show that the proposed

**Table 1: Quantitative comparison with previous UDA methods on GTA → Cityscapes. We present pre-class IoU and mIoU. The best accuracy in every column is in bold. Our results are averaged over 3 random seeds.**

| Method | Road | SW | Build | Wall | Fence | Pole | TL | TS | Veg. | Terrain | Sky | PR | Rider | Car | Truck | Bus | Train | Motor | Bike | mIoU |
|---|---|---|---|---|---|---|---|---|---|---|---|---|---|---|---|---|---|---|---|---|
| AdaptSegNet [51] | 86.5 | 36.0 | 79.9 | 23.4 | 23.3 | 23.9 | 35.2 | 14.8 | 83.4 | 33.3 | 75.6 | 58.5 | 27.6 | 73.7 | 32.5 | 35.4 | 3.9 | 30.1 | 28.1 | 42.4 |
| CyCADA [17] | 86.7 | 35.6 | 80.1 | 19.8 | 17.5 | 38.0 | 39.9 | 41.5 | 82.7 | 27.9 | 73.6 | 64.9 | 19.0 | 65.0 | 12.0 | 28.6 | 4.5 | 31.1 | 42.0 | 42.7 |
| CLAN [37] | 87.0 | 27.1 | 79.6 | 27.3 | 23.3 | 28.3 | 35.5 | 24.2 | 83.6 | 27.4 | 74.2 | 58.6 | 28.0 | 76.2 | 33.1 | 36.7 | 6.7 | 31.9 | 31.4 | 43.2 |
| SP-Adv [48] | 86.2 | 38.4 | 80.8 | 25.5 | 20.5 | 32.8 | 33.4 | 28.2 | 85.5 | 36.1 | 80.2 | 60.3 | 28.6 | 78.7 | 27.3 | 36.1 | 4.6 | 31.6 | 28.4 | 44.3 |
| MaxSquare [7] | 88.1 | 27.7 | 80.8 | 28.7 | 19.8 | 24.9 | 34.0 | 17.8 | 83.6 | 34.7 | 76.0 | 58.6 | 28.6 | 84.1 | 37.8 | 43.1 | 7.2 | 32.3 | 34.2 | 44.3 |
| ASA [80] | 89.2 | 27.8 | 81.3 | 25.3 | 22.7 | 28.7 | 36.5 | 19.6 | 83.8 | 31.4 | 77.1 | 59.2 | 29.8 | 84.3 | 33.2 | 45.6 | 16.9 | 34.5 | 30.8 | 45.1 |
| AdvEnt [54] | 89.4 | 33.1 | 81.0 | 26.6 | 26.8 | 27.2 | 33.5 | 24.7 | 83.9 | 36.7 | 78.8 | 58.7 | 30.5 | 84.8 | 38.5 | 44.5 | 1.7 | 31.6 | 32.4 | 45.5 |
| MRNet [75] | 89.1 | 23.9 | 82.2 | 19.5 | 20.1 | 33.5 | 42.2 | 39.1 | 85.3 | 33.7 | 76.4 | 60.2 | 33.7 | 86.0 | 36.1 | 43.3 | 5.9 | 22.8 | 30.8 | 45.5 |
| APODA [66] | 85.6 | 32.8 | 79.0 | 29.5 | 25.5 | 26.8 | 34.6 | 19.9 | 83.7 | 40.6 | 77.9 | 59.2 | 28.3 | 84.6 | 34.6 | 49.2 | 8.0 | 32.6 | 39.6 | 45.9 |
| CBST [81] | 91.8 | 53.5 | 80.5 | 32.7 | 21.0 | 34.0 | 28.9 | 20.4 | 83.9 | 34.2 | 80.9 | 53.1 | 24.0 | 82.7 | 30.3 | 35.9 | 16.0 | 25.9 | 42.8 | 45.9 |
| MRKLD [82] | 91.0 | 55.4 | 80.0 | 33.7 | 21.4 | 37.3 | 32.9 | 24.5 | 85.0 | 34.1 | 80.8 | 57.7 | 24.6 | 84.1 | 27.8 | 30.1 | 26.9 | 26.0 | 42.3 | 47.1 |
| FADA [56] | 91.0 | 50.6 | 86.0 | 43.4 | 29.8 | 36.8 | 43.4 | 25.0 | 86.8 | 38.3 | 87.4 | 64.0 | 38.0 | 85.2 | 31.6 | 46.1 | 6.5 | 25.4 | 37.1 | 50.1 |
| Uncertainty [76] | 90.4 | 31.2 | 85.1 | 36.9 | 25.6 | 37.5 | 48.8 | 48.5 | 85.3 | 34.8 | 81.1 | 64.4 | 36.8 | 86.3 | 34.9 | 52.2 | 1.7 | 29.0 | 44.6 | 50.3 |
| FDA [67] | 92.5 | 53.3 | 82.4 | 26.5 | 27.6 | 36.4 | 40.6 | 38.9 | 82.3 | 39.4 | 78.0 | 62.6 | 34.4 | 84.9 | 34.1 | 53.1 | 16.9 | 27.7 | 46.4 | 50.5 |
| Adaboost [77] | 90.7 | 35.9 | 85.7 | 40.1 | 27.8 | 39.0 | 49.0 | 48.4 | 85.9 | 35.1 | 85.1 | 63.1 | 34.4 | 86.8 | 38.3 | 49.5 | 0.2 | 26.5 | 45.3 | 50.9 |
| DACS [50] | 89.9 | 39.7 | 87.9 | 30.7 | 39.5 | 38.5 | 46.4 | 52.8 | 88.0 | 44.0 | 88.8 | 67.2 | 35.8 | 84.5 | 45.7 | 50.2 | 0.0 | 27.3 | 34.0 | 52.1 |
| BAPA [33] | 94.4 | 61.0 | 88.0 | 26.8 | 39.9 | 38.3 | 46.1 | 55.3 | 87.8 | 46.1 | 89.4 | 68.8 | 40.0 | 90.2 | 60.4 | 59.0 | 0.0 | 45.1 | 54.2 | 57.4 |
| ProDA [69] | 87.8 | 56.0 | 79.7 | 46.3 | 44.8 | 45.6 | 53.5 | 53.5 | 88.6 | 45.2 | 82.1 | 70.7 | 39.2 | 88.8 | 45.5 | 59.4 | 1.0 | 48.9 | 56.4 | 57.5 |
| CaCo [22] | 93.8 | 64.1 | 85.7 | 43.7 | 42.2 | 46.1 | 50.1 | 54.0 | 88.7 | 47.0 | 86.5 | 68.1 | 2.9 | 88.0 | 43.4 | 60.1 | 31.5 | 46.1 | 60.9 | 58.0 |
| DAFormer [18] | 95.7 | 70.2 | 89.4 | 53.5 | 48.1 | 49.6 | 55.8 | 59.4 | 89.9 | 47.9 | 92.5 | 72.2 | 44.7 | 92.3 | 74.5 | 78.2 | 65.1 | 55.9 | 61.8 | 68.3 |
| CAMix [79] | 96.0 | 73.1 | 89.5 | 53.9 | 50.8 | 51.7 | 58.7 | 64.9 | 90.0 | 51.2 | 92.2 | 71.8 | 44.0 | 92.8 | 78.7 | 82.3 | 70.9 | 54.1 | 64.3 | 70.0 |
| HRDA [19] | 96.4 | 74.4 | 91.0 | 61.6 | 51.5 | 57.1 | 63.9 | 69.3 | 91.3 | 48.4 | 94.2 | 79.0 | 52.9 | 93.9 | 84.1 | 85.7 | 75.9 | 63.9 | 67.5 | 73.8 |
| MIC [20] | 97.4 | 80.1 | 91.7 | 61.2 | 56.9 | 59.7 | 66.0 | 71.3 | 91.7 | 51.4 | **94.3** | 79.8 | 56.1 | 94.6 | 85.4 | **90.3** | 80.4 | 64.5 | 68.5 | 75.9 |
| CorDA† [57] | 94.7 | 63.1 | 87.6 | 30.7 | 40.6 | 40.2 | 47.8 | 51.6 | 87.6 | 47.0 | 89.7 | 66.7 | 35.9 | 90.2 | 48.9 | 57.5 | 0.0 | 39.8 | 56.0 | 56.6 |
| FAFS† [3] | 93.4 | 60.7 | 88.0 | 43.5 | 32.1 | 40.3 | 54.3 | 53.0 | 88.2 | 44.5 | 90.0 | 69.5 | 35.8 | 88.7 | 34.1 | 53.9 | 41.3 | 51.7 | 54.7 | 58.8 |
| DBST† [3] | 94.3 | 60.0 | 87.9 | 50.5 | 43.0 | 42.6 | 50.8 | 51.3 | 88.0 | 45.9 | 89.7 | 68.9 | 41.8 | 88.0 | 45.8 | 63.8 | 0.0 | 50.0 | 55.8 | 58.8 |
| Ours† | **97.5** | **80.7** | **92.1** | **66.4** | **62.3** | **63.1** | **67.7** | **75.7** | **91.8** | **52.4** | 93.9 | **81.6** | **61.8** | **94.7** | **88.3** | 90.0 | **81.2** | **65.8** | **69.6** | **77.7** |

†: Training with depth data.

method largely improved the prediction quality of challenging small object "Traffic Sign" and large category "Terrain".

**Results on Synthia→Cityscapes.** We show our results on SYN-THIA → Cityscapes in Table 1 and the results show the consistent performance improvement of our method, increasing from 67.3 to 69.3 (+2.0 mIoU) compared to the state-of-the-art method MIC [20]. Especially, our method significantly increases the IoU performance of the challenging class "SideWalk" from 50.5 to 63.1 (+12.6 mIoU). It is also noticeable that our method remains competitive in segmenting most individual classes and yields a significant increase of +6.8 on "Road", +6.6 on "Bus", +3.9 on "Pole", +3.7 on "Road", +3.2 on "Wall" and +2.9 on "Truck".

## 4.3 Ablation Study and Further Disccussion

**Ablation Study on Different Scene Adaptation Frameworks.** We combine our method with different scene adaptation architectures on GTA→Cityscapes. Table 4 shows that our method achieves consistent and significant improvements across different methods with different network architectures. Firstly, our method improves the state-of-the-art performance by +1.8 mIoU. Then we evaluate the proposed method on two strong methods based on transformer backbone, yielding +3.2 mIoU and +2.3 mIoU performance increase on DAFormer [18] and HRDA [19], respectively. Secondly, we evaluate our method on DeepLabV2 [6] architecture with ResNet-101 [16] backbone. We show that we improve the performance of the CNN-based cross-domain mixing method, *i.e.*, DACS by +4.1 mIoU. The ablation study verifies the effectiveness of our method in leveraging depth information to enhance cross-domain mixing not only on Transformer-based networks but also on CNN-based architecture.

**Ablation Study on Different Components of the Proposed Method.** In order to verify the effectiveness of our proposed components, we train four different models from M1 to M4 and show the result in Table 3. "ST Base" means the self training baseline with semantic segmentation branch and depth regression branch. "Naive Mix" denotes the cross-domain mixing strategy. "DCF" represents the proposed depth-aware mixing (Depth-guided Contextual Filter). "AFO" denotes the proposed Adaptive Feature Optimization module and we used two different method to perform AFO. Firstly, we leverage channel attention (CA) that could select useful information along the channel dimension to perform the feature optimization. In this method, the fused feature is adaptively optimized by SENet [21], the output is a weighted vector which is multiplied back to the visual and depth feature. We leavrage "AFO (CA)" to denote this method. Secondly, we leverage the iterative use of transformer block to adaptively optimize the multi-task feature. In this case, the output of the transformer block is a weighted map. The Multimodal Communication (MMC) module is then used to incorporate rich knowledge from the depth prediction. We denote this method as "AFO (Trans + MMC)". M1 is the self training baseline with depth regression based on DAFormer architecture. M2 adds the cross-domain mixing strategy for improvement and shows a competitive result of 76.0 mIoU. M3 is the model with the Depth-guided Contextual Filter, increasing the performance from 76.0 to 77.1 mIoU (+1.1 mIoU), which demonstrates the effectiveness of transferring the mixed training images to real-world layout with the help of the depth information. M4 adds the multi-task framework that leverages Channel Attention (CA) mechanism to fuse the discriminative depth feature into the visual feature. The segmentation result is increased by a small margin (+0.2 mIoU), which means CA could help the network to

**Table 2: Quantitative comparison with previous UDA methods on SYNTHIA → Cityscapes. We present pre-class IoU, mIoU and mIoU*. mIoU and mIoU* are averaged over 16 and 13 categories, respectively. The best accuracy in every column is in bold. Our results are averaged over 3 random seeds.**

| Method | Road | SW | Build | Wall* | Fence* | Pole* | TL | TS | Veg. | Sky | PR | Rider | Car | Bus | Motor | Bike | mIoU* | mIoU |
|---|---|---|---|---|---|---|---|---|---|---|---|---|---|---|---|---|---|---|
| MaxSquare [7] | 77.4 | 34.0 | 78.7 | 5.6 | 0.2 | 27.7 | 5.8 | 9.8 | 80.7 | 83.2 | 58.5 | 20.5 | 74.1 | 32.1 | 11.0 | 29.9 | 45.8 | 39.3 |
| SIBAN [36] | 82.5 | 24.0 | 79.4 | – | – | – | 16.5 | 12.7 | 79.2 | 82.8 | 58.3 | 18.0 | 79.3 | 25.3 | 17.6 | 25.9 | 46.3 | – |
| PatchAlign [52] | 82.4 | 38.0 | 78.6 | 8.7 | 0.6 | 26.0 | 3.9 | 11.1 | 75.5 | 84.6 | 53.5 | 21.6 | 71.4 | 32.6 | 19.3 | 31.7 | 46.5 | 40.0 |
| AdaptSegNet [51] | 84.3 | 42.7 | 77.5 | – | – | – | 4.7 | 7.0 | 77.9 | 82.5 | 54.3 | 21.0 | 72.3 | 32.2 | 18.9 | 32.3 | 46.7 | – |
| CLAN [37] | 81.3 | 37.0 | 80.1 | – | – | – | 16.1 | 13.7 | 78.2 | 81.5 | 53.4 | 21.2 | 73.0 | 32.9 | 22.6 | 30.7 | 47.8 | – |
| SP-Adv [48] | 84.8 | 35.8 | 78.6 | – | – | – | 6.2 | 15.6 | 80.5 | 82.0 | 66.5 | 22.7 | 74.3 | 34.1 | 19.2 | 27.3 | 48.3 | – |
| AdvEnt [54] | 85.6 | 42.2 | 79.7 | 8.7 | 0.4 | 25.9 | 5.4 | 8.1 | 80.4 | 84.1 | 57.9 | 23.8 | 73.3 | 36.4 | 14.2 | 33.0 | 48.0 | 41.2 |
| ASA [80] | 91.2 | 48.5 | 80.4 | 3.7 | 0.3 | 21.7 | 5.5 | 5.2 | 79.5 | 83.6 | 56.4 | 21.0 | 80.3 | 36.2 | 20.0 | 32.9 | 49.3 | 41.7 |
| CBST [81] | 68.0 | 29.9 | 76.3 | 10.8 | 1.4 | 33.9 | 22.8 | 29.5 | 77.6 | 78.3 | 60.6 | 28.3 | 81.6 | 23.5 | 18.8 | 39.8 | 48.9 | 42.6 |
| MRNet [75] | 82.0 | 36.5 | 80.4 | 4.2 | 0.4 | 33.7 | 18.0 | 13.4 | 81.1 | 80.8 | 61.3 | 21.7 | 84.4 | 32.4 | 14.8 | 45.7 | 50.2 | 43.2 |
| MRKLD [82] | 67.7 | 32.2 | 73.9 | 10.7 | 1.6 | 37.4 | 22.2 | 31.2 | 80.8 | 80.5 | 60.8 | 29.1 | 82.8 | 25.0 | 19.4 | 45.3 | 50.1 | 43.8 |
| CCM [29] | 79.6 | 36.4 | 80.6 | 13.3 | 0.3 | 25.5 | 22.4 | 14.9 | 81.8 | 77.4 | 56.8 | 25.9 | 80.7 | 45.3 | 29.9 | 52.0 | 52.9 | 45.2 |
| Uncertainty [76] | 87.6 | 41.9 | 83.1 | 14.7 | 1.7 | 36.2 | 31.3 | 19.9 | 81.6 | 80.6 | 63.0 | 21.8 | 86.2 | 40.7 | 23.6 | 53.1 | 54.9 | 47.9 |
| BL [31] | 86.0 | 46.7 | 80.3 | – | – | – | 14.1 | 11.6 | 79.2 | 81.3 | 54.1 | 27.9 | 73.7 | 42.2 | 25.7 | 45.3 | 51.4 | – |
| DT [59] | 83.0 | 44.0 | 80.3 | – | – | – | 17.1 | 15.8 | 80.5 | 81.8 | 59.9 | 33.1 | 70.2 | 37.3 | 28.5 | 45.8 | 52.1 | – |
| IAST [38] | 81.9 | 41.5 | 83.3 | 17.7 | 4.6 | 32.3 | 30.9 | 28.8 | 83.4 | 85.0 | 65.5 | 30.8 | 86.5 | 38.2 | 33.1 | 52.7 | 49.8 | - |
| DAFormer [18] | 84.5 | 40.7 | 88.4 | 41.5 | 6.5 | 50.0 | 55.0 | 54.6 | 86.0 | 89.8 | 73.2 | 48.2 | 87.2 | 53.2 | 53.9 | 61.7 | 67.4 | 60.9 |
| CAMix [79] | 87.4 | 47.5 | 88.8 | – | – | – | 55.2 | 55.4 | 87.0 | 91.7 | 72.0 | 49.3 | 86.9 | 57.0 | 57.5 | 63.6 | 69.2 | – |
| HRDA [19] | 85.2 | 47.7 | 88.8 | 49.5 | 4.8 | 57.2 | 65.7 | 60.9 | 85.3 | 92.9 | 79.4 | 52.8 | 89.0 | 64.7 | 63.9 | 64.9 | 72.4 | 65.8 |
| MIC [20] | 86.6 | 50.5 | 89.3 | 47.9 | 7.8 | 59.4 | 66.7 | 63.4 | 87.1 | **94.6** | **81.0** | **58.9** | 90.1 | 61.9 | **67.1** | 64.3 | 74.0 | 67.3 |
| DADA [55] | 89.2 | 44.8 | 81.4 | 6.8 | 0.3 | 26.2 | 8.6 | 11.1 | 81.8 | 84.0 | 54.7 | 19.3 | 79.7 | 40.7 | 14.0 | 38.8 | 49.8 | 42.6 |
| CorDA† [57] | 93.3 | 61.6 | 85.3 | 19.6 | 5.1 | 37.8 | 36.6 | 42.8 | 84.9 | 90.4 | 69.7 | 41.8 | 85.6 | 38.4 | 32.6 | 53.9 | 62.8 | 55.0 |
| Ours† | **93.4** | **63.1** | **89.8** | **51.1** | **9.1** | **61.4** | 66.9 | 64.0 | 88.0 | 94.5 | 80.9 | 56.6 | **90.9** | **68.5** | 63.7 | **66.6** | **75.9** | **69.3** |

†: Training with depth data.

**Table 3: Ablation study of different components of our proposed framework on GTA→Cityscapes. The results are averaged over 3 random seeds.**

| Method | ST Base. | Naive Mix. | DCF. | AFO. (CA) | AFO. (Trans + MMC) | mIoU↑ |
|---|---|---|---|---|---|---|
| M1 | ✓ | | | | | 73.1 |
| M2 | ✓ | ✓ | | | | 76.0 |
| M3 | ✓ | ✓ | ✓ | | | 77.1 |
| M4 | ✓ | ✓ | ✓ | ✓ | | 77.3 |
| M5 | ✓ | ✓ | ✓ | | ✓ | 77.7 |

**Table 4: Compatibility of the proposed method on different UDA methods and backbones on GTA→Cityscapes. Our results are averaged over 3 random seeds.**

| Backbone | UDA Method | w/o | w/ | Diff. |
|---|---|---|---|---|
| DeepLabV2 [6] | DACS [50] | 52.1 | 56.2 | +4.1 |
| DAFormer [18] | DAFormer [18] | 68.3 | 71.5 | +3.2 |
| DAFormer [18] | HRDA [19] | 73.8 | 76.1 | +2.3 |
| DAFormer [18] | MIC [20] | 75.9 | 77.7 | +1.8 |

adaptively learn to focus or to ignore information from the auxiliary task to some extent. M5 is our proposed depth-aware multi-task model with both Depth-guided Contextual Filter and Adaptive Feature Optimization (AFO) module. Compared to M3, M5 has a mIoU increase of +0.6 from 77.1 to 77.7, which shows the effectiveness of multi-modal feature optimization using transformers to facilitate contextual learning.

**Ablation study on GTA+SYNTHIA → Cityscapes.** We evaluate the proposed method on multi-source domains setting and report the quantitative result on GTA+SYNTHIA → Cityscapes. With multi-source domain data, the model can be trained more robust to the unlabelled target environment. We adopt DACS [50] as our baseline

**Table 5: Quantitative results on GTA+SYNTHIA → Cityscapes. The performance is provided as mIoU in %.**

| Baseline (Single Source) | Multi Source | Multi Source + Depth |
|---|---|---|
| 52.1 | 54.2 | 56.7 |

with 52.1 mIoU (Only GTA) performance shown in Table 5. With more source-domain data, the model yields a better result of 54.2 mIoU. Then, we can observe that our method yields a larger improvement from 54.2 to 56.7 mIoU, demonstrating that the proposed model could adapt multi-domain depth to the target domain and hence increase performance.

## 5 CONCLUSION

In this work, we introduce a new depth-aware scene adaptation framework that effectively leverages the guidance of depth to enhance data augmentation and contextual learning. The proposed framework not only explicitly refines the cross-domain mixing by stimulating real-world layouts with the guidance of depth distributions of objects, but also introduced a cross-task encoder that adaptively optimizes the multi-task feature and focused on the discriminative depth feature to help contextual learning. By integrating our depth-aware framework into existing self-training methods based on either transformer or CNN, we achieve state-of-the-art performance on two widely used benchmarks and a significant improvement on small-scale categories. Extensive experimental results verify our motivation to transfer the training images to real-world layouts and demonstrate the effectiveness of our multi-task framework in improving scene adaptation performance.

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
