# OpenReview forum: "Transferring to Real-World Layouts: A Depth-aware Framework for Scene Adaptation"
_acmmm.org/ACMMM/2024/Conference — MM2024 Oral_

### Official Review · Reviewer_tEjP · 2024-05-24

**Rating:** 6
**Confidence:** 4

**Summary:**

This paper proposes a depth-aware unsupervised domain adaptation framework for semantic segmentation tasks. The method utilizes a Depth-Guided Contextual Filter (DCF) and a cross-task encoder to enhance the mixing of source and target domain data using depth information, improving segmentation performance for small objects and complex scenes. Experimental results show that this method achieves significant performance improvements on the GTA→Cityscapes and Synthia→Cityscapes datasets. Even when using pseudo depth data, the method achieves competitive performance.

**Strengths:**

1.The proposed depth-aware framework in the article represents an innovative approach that utilizes depth estimation to clearly distinguish and blend semantic categories, thereby simulating the layout of the real world. This method effectively addresses the issue of layout mismatch that may arise from simply blending source and target domain data in traditional unsupervised domain adaptation methods.
2.By introducing a cross-task encoder, the framework achieves contextual learning and feature fusion for the two complementary tasks of segmentation and depth learning. This collaborative learning approach across tasks contributes to extracting more comprehensive and accurate feature representations, thereby enhancing the performance of scene segmentation.

**Limitations:**

From what I know, the author has used all the datasets related to domain adaptation tasks from virtual scenes to real scenes in autonomous driving. The experiments compared many models related to previous research in this field, but is it necessary to include some models that are slightly old in time and have significantly different effects in the comparison list? At the same time, I think the method proposed by the author is universal and can also be tested in some tasks in  adverse scenes, such as Cityscapes to ACDC, for example.

**Suitability:**

2

---

### Official Review · Reviewer_fR8C · 2024-05-29

**Rating:** 4
**Confidence:** 4

**Summary:**

This paper proposes a novel framework to improve scene segmentation via unsupervised domain adaptation (UDA), which transfers knowledge from synthetic data to real-world data. The framework utilizes depth estimation to refine the mixing of data across domains, aiming to create more realistic training samples. A key component is the Depth-guided Contextual Filter (DCF), which adjusts the mixing based on depth, improving the relevance to real-world layouts. Even using pseudo depth data, The method achieves competitive results on standard benchmarks, demonstrating its effectiveness in reducing manual annotation needs and facilitating better domain adaptation.

**Strengths:**

The analysis is comprehensive and well-developed.
The paper is well-written and easy to understand.
The experimental part evaluates a rich variety of datasets, and the results are convincing. The ablation experiments also demonstrate the role of key modules.

**Limitations:**

1. Limited improvement: The joint use of the DCF and AFO modules proposed in this paper only shows a 1.7% improvement over the Naive Mix, but significantly increases the computation resources and time required in the sample preprocessing stage.
2. From the ablation study in Table 3, the main improvement comes from the DCF module, while the elaborately designed multimodal communication AFO module has a very limited effect. The introduction of the transformer block in the AFO module to establish associations between two modalities significantly increases the number of parameters, complicating the training process and extending training time, yet results in only a very limited improvement (0.2% to 0.6%).
3. The paper focuses solely on cross-domain semantic segmentation tasks, but the AFO module needs to optimize losses for both tasks (semantic segmentation and depth estimation), requiring supervision with both semantic and depth ground truths. This significantly increases the difficulty of data acquisition and annotation in real-world applications.
4. The joint optimization of losses for depth estimation and semantic segmentation tasks increases the difficulty of optimization, making the model harder to learn. While it is reasonable to assume that depth features could aid the performance of semantic segmentation tasks, joint optimization is not necessarily the best approach. Generally, optimizing multiple tasks simultaneously can lead to decreased performance on individual tasks.

**Suitability:**

2

---

### Official Review · Reviewer_7MaS · 2024-06-01

**Rating:** 4
**Confidence:** 2

**Summary:**

This paper focuses on unsupervised domain adaptation for scene segmentation. In particular, it highlights that layouts are important in real-world scenarios, and proposes a depth-aware framework to explicitly leverage depth estimation to mix the categories. The proposed DCF simulates real-world layouts and further adaptively fuses the complementing features via the cross-task encoder. Massive experimental results demonstrate its effectiveness.

**Strengths:**

1. The motivation and proposed method is reasonable. Introducing depth into domain adaptation is practical in real-world scenarios.
2. The paper is well-organized and easy to follow.

**Limitations:**

1. Since you use depth maps as supervision (which is easy to obtain on synthetic), I wonder whether the method can be generalized on real-to-real datasets. Can you try on Cityscape -> ACDC?
2. The proposed method introduces depth as supervision, can you also try to apply such designs on MIC?
3. The provided qualitative visualization shows some improvement in terrain, missing largely improved categories such as pole and wall show (refer to Table 1). Can you add more qualitative results for the significantly improved categories?

**Suitability:**

2

---

### Meta-Review · Area_Chair_nYZT · 2024-07-02

**Recommendation:** Accept (Oral)
**Confidence:** 5

**Metareview:**

The paper studies the problem of incorporating depth for unsupervised semantic scene adaptation. The advantages include: 1. The paper presents a reasonable and practical depth-aware framework for domain adaptation in real-world scenarios. 2. The well-organized and comprehensive analysis is easy to follow and understand. 3. A variety of datasets and convincing results, along with ablation experiments, support the proposed method. 4. The innovative approach uses depth estimation to address layout mismatch issues in traditional unsupervised domain adaptation methods. 5. A cross-task encoder enables contextual learning and feature fusion, improving scene segmentation performance through collaborative learning across tasks. The disadvantages include: 1. The marginal improvement in the joint use of the DCF and AFO modules. 2. Results on real to real adaptation tasks.

Overall, the paper has a high quality. All reviewers agree to accept this work. The ACs do not find significant reasons to overturn reviewers' recommendations and thus suggest accepting this paper.